# Ferrets (*Mustela furo*) Are Aware of Their Dimensions

**DOI:** 10.3390/ani13030444

**Published:** 2023-01-28

**Authors:** Ivan A. Khvatov, Alexey Yu. Sokolov, Alexander N. Kharitonov

**Affiliations:** 1Center for Biopsychological Studies, Moscow Institute of Psychoanalysis, 121170 Moscow, Russia; 2Institute of Psychology, Russian Academy of Sciences, 129366 Moscow, Russia

**Keywords:** self-awareness, body awareness, body size awareness, ferrets

## Abstract

**Simple Summary:**

Self-awareness is defined as the ability of an individual to perceive one’s physical and mental properties separately from the characteristics of the external world and/or other individuals. Traditionally, self-awareness is considered as one of the attributes of human consciousness, indicating the meaningfulness of human behavior. In recent years, there has been increasing evidence of self-awareness in different animal species. One of the earliest evolutionary components of self-awareness is body size awareness, which is expressed in the ability to consider the boundaries of one’s own body when interacting with environmental objects. In this study, we examined body size awareness in ferrets. During the experiments, the animals had to pass through holes, some of which were too small for their bodies to penetrate, and some were suitable. The results show that ferrets could pre-select the penetrable opening, even when the impassable ones were larger. We believe that these data indicate the presence of body size awareness traits in ferrets, which in turn suggests their self-awareness. This opens the way for the study and discussion of self-awareness in many other animals.

**Abstract:**

Self-awareness is a complex phenomenon expressed as the ability of an individual to separate “self-entity” from “other entity”. One of its earliest evolutionary components is body size awareness, namely, the ability to consider the boundaries of one’s own body as factors influencing interaction with surrounding objects. For ferrets, *Mustela furo*, the task requiring the penetration of various holes is ecologically relevant. We designed an experimental study in which the ferrets were supposed to select one opening out of three to get the bait. The first experiment was aimed at studying whether ferrets would prefer the holes basing on the hole size. In the second experiment, we tested the ferrets’ ability to select a single passable hole on the first try while the impassable ones were larger in area. Results from the first experiment show that when choosing from the three passable openings, the animals preferred the shortest path to the bait and ignored the size of the holes. In the second experiment, all tested ferrets preferred to penetrate the passable opening on the first attempt, even though the areas of the two impenetrable ones were larger. We argue that these data indicate that ferrets are aware of their own body size.

## 1. Introduction

This study focused on body awareness—the ability of an individual to identify possible physical interactions of their body with objects in the environment [1]. This phenomenon contributes to the survival of most species [2,3]. Two forms of body awareness are often studied experimentally: body weight and body size awareness. Body weight awareness has been demonstrated in children (*Homo sapiens*) aged 18–26 months [4], Asian elephants (*Elephas maximus*) [1], domesticated dogs (*Canis familiaris*) [5], and brown rats (*Rattus norvegicus*) [6]. Body size awareness is the ability to consider the boundaries of one’s own body when interacting with environmental objects [7]. This ability has been demonstrated in children aged 18–26 months [4], domesticated dogs [8], budgerigars (*Melopsittacus undulatus*) [9], and hooded crows (*Corvus cornix*) [7]. Our study extends these efforts by assessing body size awareness in ferrets (*Mustela furo*). Feral ferrets live in dens, such as old rabbit holes and caves under rocks and tree roots [10]. Domesticated ferrets prefer to hide in relatively inaccessible places [11], and they also prefer to enter small and narrow holes, pipes, etc. [10]. Ferrets have been used to run wire through narrow tubing in aircraft [12]. Thus, it is an ecological task for them to pass through holes of various diameters.

A variety of approaches have been used to assess body size awareness; most require the tested animals to choose an appropriately sized hole that allows them to pass from one side of a partition to another. In these experiments, the sizes of the holes varied to include holes that were too small to allow passage of the subject [4,7,8,9]. The test system used in this article is an adaptation of the method used to assess body size awareness in crows [7]. Six hooded crows participated in that study. The two spaces in the experimental setup were separated by a partition with three holes in it. Two types of experiments were conducted. To establish whether crows prefer holes based on their size, all holes were passable, but their size and location were different. The second experiment included two types of trials: background and test. In the background trials, two large passable holes and one impassable and smaller hole were used. In the test trials, one passable hole and two impassable but large holes were used. The experimental series was built in such a way that the test trial followed two background trials, provided that the location of the holes in each trial varied quasi-randomly to exclude the possibility of learning. In the second experiment, the ability of the birds to unmistakably choose a suitable hole for passing was revealed, even if the impassable holes were larger.

Members of the mustelid family are highly intelligent and adaptable, and therefore have a wide range of habitats, including urban environments [13]. In particular, ferrets attract the attention of researchers because of their mobility, playfulness, and natural exploration instinct; therefore, they are often used as model animals for studying neural development, visual [14] and auditory function [15], sexual behavior [16], hunting behavior [17], nutrition [18], toxicology [19], pharmacology [19,20], and endocrinology [19]. The ability of ferrets to learn in a maze has been established [21,22], although some researchers have noted that it occurs with difficulty due to the desire of animals to explore every dead end on the way to the goal, even following long food deprivation periods [23]. Numerical competence has also been demonstrated in ferrets [24]. Ferrets have also been studied for self-recognition in mirrors [25], and demonstrated promising results in mirror preference and mark tests. 

The present study was dedicated to the development of representations of size and boundaries of one’s own body in ferrets.

## 2. Materials and Methods

### 2.1. Subjects

Six male ferrets (*Mustela furo*), aged 2–3 years, took part in the experiments. None were excluded from the original sample during the experiment. We considered the sample size sufficient for obtaining reliable statistical data, since six individuals were used in experiments on crows using a similar method [7]. Similarly, six individuals were used as part of the ferrets MSR study [25]. The animals were raised and kept in a vivarium at the Center for Biopsychological Research (Moscow Institute of Psychoanalysis, Moscow, Russia). The experiments were conducted in conformity with standing standards for animal research (also see the Institutional Review Board Statement below).

### 2.2. Experimental Setup

The experimental setup (Figure 1) consisted of a rectangular arena (110 × 130 × 40 cm). It was divided into two halves by a partition (110 × 40 cm), in which there were three holes. Additional plates were inserted into special grooves located on the sides of these holes, which made it possible to adjust their size and shape or completely close the holes. In the middle of the arena, there was a partition with three 30 × 30 cm holes. A launch box (40 × 40 × 40 cm) equipped with a remote-controlled door was attached to one of them. Attached to another hole in the wall was a finish box of the same size, where the bait for the tested animal was placed.

The walls were made of transparent plexiglass; the central partition panel and the plates placed within it, which controlled the size and shape of the holes, were opaque.

Video recording was carried out using a video camera Panasonic HC-V260 (Panasonic Holdings Corp., manufactured in Selangor, Malaysia) located behind the starting box at a height of 1.5 m and directed towards the rectangular arena (Figure 2). Moreover, for remote monitoring of the progress of the experimental trials, a video surveillance system was used.

### 2.3. General Procedure

The animals were deprived of food for one hour before each training or testing session.

Before the start of the experimental session, the animal was placed in the starting box. A closed starting box was placed in front of the entrance to the arena, after which the experimenter left the room with the experimental setup. The experimenter opened the door of the launch box and monitored the behavior of the ferret remotely.

The test was considered completed if the ferret passed through one of the holes in the partition and entered the finishing box with the bait in five minutes or less. If the animal did not pass through the hole within five minutes, the experiment was terminated.

Two experiments were conducted, which were preceded by training. A total of 117 trials were performed with each animal: 9 in the training, 36 in the first experiment, and 72 in the second experiment. Each animal participated in 6–9 trials per day. Thus, each ferret was engaged from 13 to 18 days.

### 2.4. Training

The purpose of the training was to familiarize the ferrets with the experimental setup and the experimental procedure.

During preliminary observations, we found that the smallest hole for a ferret to penetrate had a height and width of slightly over 60 mm (or the same diameter if round). Based on this, we set the minimum size of the passable opening to 70 mm.

Initially, the animals simply had to go through one of the three openings in the partition and enter the finishing box to get the bait. Only one hole (30 × 30 cm) was opened during each trial, the location of which varied quasi-randomly so that the same hole was open in no more than two trials in a row, and each hole was opened the same number of times (three times at each position: left, right, and center). The criterion for completing training and moving on to the experimental sessions was the number of trials. By the ninth trial, the time to completion was less than 20 s; in subsequent trials, this indicator did not decrease. Therefore, each ferret performed nine trials.

### 2.5. Experiment 1

The purpose of this experiment was to learn whether the ferrets would try to pass through a larger hole provided that two other holes of smaller areas were also passable.

In each trial, the partition bore three holes of different diameters: small (Ø 70 mm), medium (Ø 90 mm), and large (Ø 110 mm). The location of the holes in the samples varied quasi-randomly: in each subsequent sample, a certain diameter was in a different position (left, central or right), a hole of each diameter was applied in all positions an equal number of times throughout the experiment (Figure 3). As the experimental session consisted of 36 trials, the holes of each diameter were in the left, central, and right positions 12 times. We considered this number of trials sufficient for identifying the preferred holes, since the same number was used in the experiment on crows using a similar method [7].

#### Statistical Analysis

In each trial, two indicators were registered: when the ferret first approached the hole and the passage through the hole. A situation where the distance between the tip of the muzzle and the hole was no more than 10 cm was considered “an approach”.

The data were processed using STATISTICA (data analysis software system), version 10 (version number 10.0.1011.0), StatSoft Russia (Moscow, Russia).

To assess the frequency of connections between the first approach followed by passage through the opening, we compared the empirical distribution of the total number of approaches with subsequent passage with the number of approaches after which the animals did not enter the opening, but went to another opening with a hypothetical uniform distribution (50%/50%). For such a comparison, we applied Pearson’s chi-square test (χ^2^).

Factorial ANOVA (*n* = 6) was used to identify predictors of hole selection. The following were used as predictors: location (left/center/right) and hole area (small/medium/large). The dependent variable was the number of passes through the holes. Two-way relationships between predictors were also analyzed. Subject ID was included as a random factor. We also applied Tukey’s test to detect the effect of differences between predictor levels.

### 2.6. Experiment 2

The purpose of this experiment was to determine whether ferrets would choose one passable hole out of three when the other two holes were larger in overall area but impassable.

Two types of conditions were used in the experiment: test and background.

Each animal participated in 24 test trials. In each test trial, only one hole was passable, though it had a significantly smaller area than the other two impassable ones.

As we had established previously, a Ø 70 mm circular opening or a square with the same side length were the smallest passable openings. Thus, the rectangular openings with either height or width less than 70 mm would be impassable even if they had a larger area. Accordingly, we used two types of large but impassable holes. The first type was a vertical rectangle 25 × 250 mm: its area was 1.6 times the size of the smallest passable hole, whereas the width was only 0.4 times. The second type was a horizontal rectangle 250 × 25 mm: its area was 1.6 times the size of the smallest passable hole, whereas the height was 0.4 times the height of the smallest passable hole.

Small passable holes of two types were used. The first type was a circle with a diameter of 70 mm. The second type was a square with a 70 mm side.

The types and arrangement of passable and impassable holes varied quasi-randomly so that two conditions were met. First, the same number of passable and impassable holes of both types were used during 24 trials. Second, the passable and impassable holes of different types were combined and appeared an equal number of times across all 24 trials (Figure 4). The purpose of this alternation was to eliminate the effect of learning to penetrate a hole of a certain shape.

The animals participated in 48 background trials. We used two passable holes of a larger diameter and one impassable hole of a smaller diameter in each background trial.

The impassable holes were also of two types. The first type was a Ø 40 mm circle, that is, it was 0.6 times the size of the minimum passable diameter. The second type was a square with a side of 40 mm, that is, its width and height were 0.6 times the size of the width and height of the minimum passable square.

The passable openings that were used had a larger area than the impassable ones, and in one of the measurements (height or width), they corresponded to the minimum passable hole. Two types of such holes were used. The first type was a vertical rectangle 70 × 250 mm. The second type was a horizontal rectangle 250 × 70 mm.

The types of passable and impassable holes in the background trials varied in the same way as in the test samples (Figure 4).

The test and background trials were alternated in such a way that two background trials followed one test trial (therefore, there were twice as many background trials as test trials).

In addition, the method we used earlier to study body size awareness in crows was modified [7]. In the crow study, 12 test trials (with one smaller passable hole) and 24 background trials (with two larger passable holes) were arranged. In the test trials, a square hole was always permeable. Thus, there remained the possibility that the animal could learn to penetrate holes of a certain shape. In the ferret experiment, to exclude this possibility in the test trials, we used passable holes of two shapes (round and square), alternating them. For this reason, the total number of test and background trials turned out to be twice as large as in the experiment on crows (two background trials were conducted between two test trials). We left the ratio of the number of test and background samples the same as in the experiment with crows [7], since in the experiment on birds it turned out to be effective in identifying the preferred holes for penetration.

#### Statistical Analysis

The results of 24 test trials were analyzed. In each test trial, two indicators were considered: when the ferret first approached the hole and the first attempt to pass through the hole. A disposition was qualified “an approach” when the distance between the tip of the muzzle of the ferret and the hole was no more than 10 cm. A situation was considered “an attempt” when at least the tip of the muzzle (the nose of the ferret) was inserted into the hole. In this case, we considered this the penetration attempt, since in the case of an impenetrable hole, full penetration could not be carried out. We analyzed these indicators since they indicated the selection of a hole without prior physical contact with it.

Using the binomial criterion individually for each animal, we estimated the significance of the difference in the number of first approaches relating to the passable hole and first attempts to penetrate this hole from the random level (33.3). The effect size was measured using Cohen’s “d” [26].

In order to reveal the relationship between the first approach relating to a certain hole and the attempt of subsequent penetration, we compared the empirical distribution of the number of approaches, after which the ferret made an attempt to penetrate them, and the cases when the ferret tried to enter another hole, with a hypothetical uniform distribution (50%/50%). For comparison, we used Pearson’s chi-square test (χ^2^).

Factorial ANOVA (*n* = 6) was applied to identify the predictors of hole selection in 24 test trials. The predictors were hole passability (passable/impassable), hole position (right/center/left), impassable hole orientation (vertical/horizontal), and hole shape (round/square). The number of first approaches and the number of first attempts to pass through the holes were the dependent variables. Additionally, we considered the two-way interactions between predictors. Subject ID was included as a random factor. We also applied Tukey’s test to detect the effect of differences between predictor levels.

A similar factorial ANOVA (*n* = 6) was applied to identify the predictors of hole selection in 48 background trials.

Using the Friedman test, it was determined whether the number of attempts to pass through inappropriate and suitable holes changed during 24 test trials.

## 3. Results

### 3.1. Experiment 1

The pooled data for all 6 ferrets show that the animals were significantly more likely to pass through the same hole they approached first in 203 out of 216 cases (χ^2^ = 103.606; df = 1; *p* < 0.001).

The results of factorial ANOVA show that the only predictor determining the choice of hole for passing was the position of the hole. Ferrets were significantly more likely to pass through the middle hole than through the left or right ones, regardless of its size (F_(2, 45)_ = 165.36; *p* < 0.001; Tukey’s test, *p* = 0.0001). No interaction between predictors was revealed (Table 1; Figure 5).

### 3.2. Experiment 2

The total data for all 6 ferrets, as in the previous experiment, show that in 24 test trials, animals significantly more often passed through the hole they approached first (137 of 144 cases) χ^2^ = 73.696; df = 1; *p* < 0.001.

Frequently, ferrets that first approached a passable hole then immediately penetrated the same hole. Of 24 trials, 3 ferrets made their first attempt immediately after their first approach in relation to a passable but small hole on 22 occasions (22/24; *p* < 0.001; binomial test), 1 ferret did so in 23 trials (23/24; *p* < 0.001; binomial test), and 2 more did so in all 24 trials (24/24; *p* < 0.001; binomial test).

We have found that the effect size for the number of first approaches in relation to passable openings, which were immediately followed by penetration of the hole, was high for six subjects (Cohen’s d = 15.13; *n* = 6; M = 22.83, SD = 0.98).

Below are the results of a separate analysis of the first penetration attempts in a trial made by ferrets into the passable hole. Of the 24 samples, 1 ferret made 22 first attempts to penetrate the passable hole (22/24; *p* < 0.001; binomial test), 2 ferrets made 23 first attempts (23/24; *p* < 0.001; binomial test), and 3 more ferrets made the first attempt to penetrate the passable hole in all trials (24/24; *p* < 0.001; binomial test).

We found that the effect size for the number of first attempts to penetrate passable holes in the sample, for six subjects, was high (Cohen’s d = 18.69; *n* = 6; M = 23.33, SD = 0.82).

Factorial ANOVA results for 24 test trials show that hole passability was the only predictor of both first trial hole approach (F_(1, 129)_ = 1781.8; *p* < 0.001) (Table 1; Figure 6b) and first attempt to pass through the hole (F_(1, 129)_ = 2312.0; *p* < 0.001) (Table 1; Figure 6a). The ferrets significantly more often first approached the passable holes in a trial (Tukey’s test, *p* = 0.00002), and the first attempt of penetration of the passable hole (Tukey’s test, *p* = 0.00002), regardless of the shape of the passable holes, location of holes, or orientation of the impassable but large holes.

Over 24 trials, the number of attempts to penetrate the impassable holes did not change significantly for all 6 ferrets (Friedman test = 20.4, *n* = 6, cc = 23, *p* = 0.615).

During the background tests, the ferrets in all cases made attempts to pass through the hole that they first approached, without a single attempt to penetrate the impenetrable hole, i.e., they did not make any mistake. Therefore, when performing factorial ANOVA, the number of first approaches in the sample was considered as a dependent variable. Hole passability (F(1, 129) = 1889.1; *p* < 0.001), hole position (F(1, 129) = 258.3; *p* < 0.001), and the interaction of these two factors (F(1, 129) = 258.3; *p* < 0.001) affected the choice. Ferrets were significantly more likely to make the first approach in a trial relating to passable holes than to unpassable ones (Tukey’s test, *p* = 0.00002). Furthermore, they were more likely to first approach the central holes than the left or right ones (Tukey’s test, *p* = 0.00002).

## 4. Discussion

### 4.1. Experiment 1

The results of the first experiment indicate that the hole size did not influence the selection of the hole for passing. The determining and only predictor of choice was the position of the opening: ferrets significantly more often preferred to penetrate the central holes. We believe that this is because the shortest path to the bait ran through the central hole, which was located directly opposite the starting point, and through which the bait might have been seen when the animal approached the hole.

In the first experiment, when passable holes were of all three diameters (Ø 70 mm, Ø 90 mm, and Ø 110 mm), the ferrets had to perform a similar number of motor acts to pass through large, medium, and small holes; thus, the position of the hole was the only predictor of choice. This may be because the task that requires penetration of holes is more ecologically relevant for ferrets [12,19] than, for example, for crows [7].

### 4.2. Experiment 2

The results of the second experiment show that the passability of the hole was the only factor determining both the choice of the first approach and the first attempt to penetrate the hole in a trial. At the same time, ferrets not only significantly more often made the first approach in relation to the passable opening in the sample, but in the overwhelming majority of cases, immediately after that, they also penetrated this hole, as evidenced by the analysis carried out individually for each animal. In total, for all 24 tests per one animal, all 6 ferrets made only 7 mistakes (i.e., failed to make the first approach and subsequent penetration of the passable hole) out of total 144 cases.

The possibility that ferrets quickly learned a same–different rule and chose the different passage in test trials should also be explored. However, in the first test trial, all six ferrets first directly approached the passable hole and immediately penetrated it. In addition, statistical analysis showed that the number of unsuccessful attempts to pass through the impassable but large holes over 24 test trials did not change significantly. This is indicative of the absence of learning effect.

We believe that these data should be interpreted as evidence that the ferrets, having the intention to reach the bait by overcoming the barrier, in advance (even before physical contact) chose the holes suitable for passing and then passed through them. In turn, this suggests that ferrets, having an idea of their own body size, could match it with the size of the opening, which is a sign of body size awareness.

### 4.3. General

A number of studies have yielded data on body size awareness in mammals [4,8] and birds [7,9].

In a study on hooded crows [7], almost literally replicated in the present study, the holes of square and rectangular shape differed in size, and the rectangular ones could be placed either in vertical or horizontal position. Under these conditions, the birds significantly more often passed through the passable holes that they approached first. However, they approached the rectangular passable hole first only if it was higher than the larger but unsuitable for passage hole. If the passable hole was lower than the impassable but larger hole, the birds more often first approached the impassable hole, although they usually did not attempt to penetrate. If the passable holes differed only in width, the birds chose the wider one, and they generally preferred the higher openings since in order to penetrate the passable but low hole, the crows had to crouch and make other obviously inconvenient additional movements. Similar data were obtained in dogs: the animals approached low holes more slowly [8]. 

Accordingly, ferrets perform better than hooded crows (they make fewer mistakes), suggesting a better development of body size awareness. However, this may be because the task of relating their size to the size of the holes is ecologically more relevant for ferrets, i.e., they are more likely to face similar tasks in their natural habitat. In the latter case, the results of this study would further imply the need for cross-species comparisons. However, in our opinion, strict replication of research methods, assessment criteria, etc., is hardly possible because of difference in anatomy, physiology, behavior, ecology, and numerous other aspects of life of different species. Given this, we stick to a broader criterion of comparison based on behavioral relevance of the experimental task: one cannot expect a sessile animal to penetrate holes. In any case, no consideration may change the fact that ferrets are smart in hole penetration tasks, thus demonstrating body size awareness to the extent of the adequacy of the method used.

In a body size awareness study conducted on human children aged 18 to 26 months, the latter also had to choose between two holes in a septum: passable and impassable. It was found that at the age of 18 months, children chose the correct hole only after they made one mistake or more in one trial; by the 22nd month, the number of errors decreased, and only 26-month-old children did not make mistakes and chose the passable hole immediately [4]. As in the previously mentioned and subsequently cited works, the results of the studies on children cannot be directly compared with the results of our work due to differences between the methods. However, one may argue that the ferrets are comparable in body size awareness with the 26-month-old children.

In experiments aimed at studying body size awareness conducted on dogs [8] and parrots [9], the subjects were required to overcome the partition in the experimental setup by penetrating the only hole, and the sizes of the holes varied. Therefore, these animals, unlike the ferrets in our experiment, did not need to choose between holes. The dependent variable in dogs [8] was the latent period preceding the passage of the dogs or the folding of wings by flying parrots to penetrate a hole [9]. This difference in the criteria for assessing body size awareness makes it difficult to directly compare the results of these two studies with the results discussed in this article.

We should stress the importance of further studies of evolutionary diversity of the phenomenon of body awareness using different variations of the described method, specifically, in non-mammal species. In particular, we used a technique that stimulates an animal to pass through various openings, and found that the radiated ratsnake (*Elaphe radiata*) cannot mentally compare the size of their body with the size of the hole, but they can solve this problem after dozens of trials, that is, with the help of learning [27]. Studies on terrestrial hermit crabs (*Coenobita compressus*) have shown that these invertebrates can consider the size of their body and change in shell structure [28,29].

## 5. Conclusions

Our experiments show that ferrets can consider the size of their own body, correlating it with the size of holes even before they come into direct physical contact. This allows us to state that ferrets manifest traits of body size awareness. Body size awareness is probably one of the most evolutionarily ancient components of self-awareness [3]. Moreover, it can be assumed that self-awareness is a modular phenomenon, that is, it may consist of separate mechanisms (for example, olfactory awareness [30,31,32], weight-related self-awareness [1,4,5], size-related self-awareness [4,7,8,9], awareness of own appearance [33,34]) formed at different stages of evolution. At the same time, individual modules can be better developed in certain animal species due to the peculiarities of their behavior and living conditions. Verification of this hypothesis may be the objective of further research.

Thus, this work included, it has now been established that many animal species, i.e., humans after 22–26 months of age [4], dogs [8], budgerigars [9], hooded crows [7], and ferrets manifest body size awareness. Attempts to demonstrate this cognitive ability in reptiles, such as ratsnakes [27], and even invertebrates [28,29] provide a new challenge to understanding body size awareness and may signal different evolutionary roots of the phenomenon in different species.

## Figures and Tables

**Figure 1 animals-13-00444-f001:**
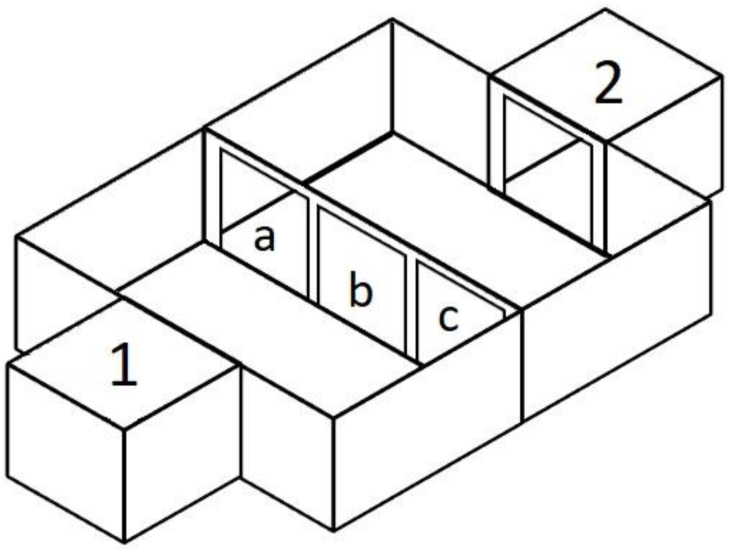
The experimental setup: 1—launch box; 2—finish box with bait; a, b, c—holes in the partition.

**Figure 2 animals-13-00444-f002:**
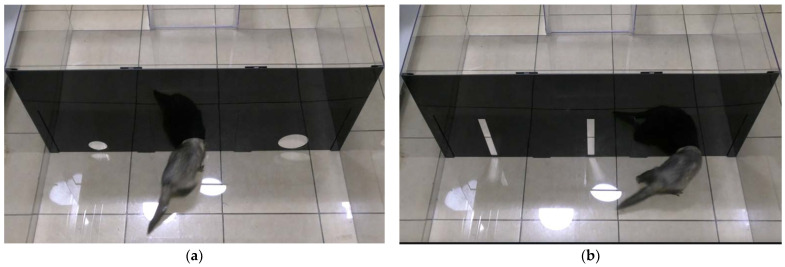
View of the experimental setup with ferrets: (**a**) a ferret enters the hole during Experiment 1; (**b**) a ferret enters a hole during Experiment 2. (Photo: I.A.Khvatov).

**Figure 3 animals-13-00444-f003:**
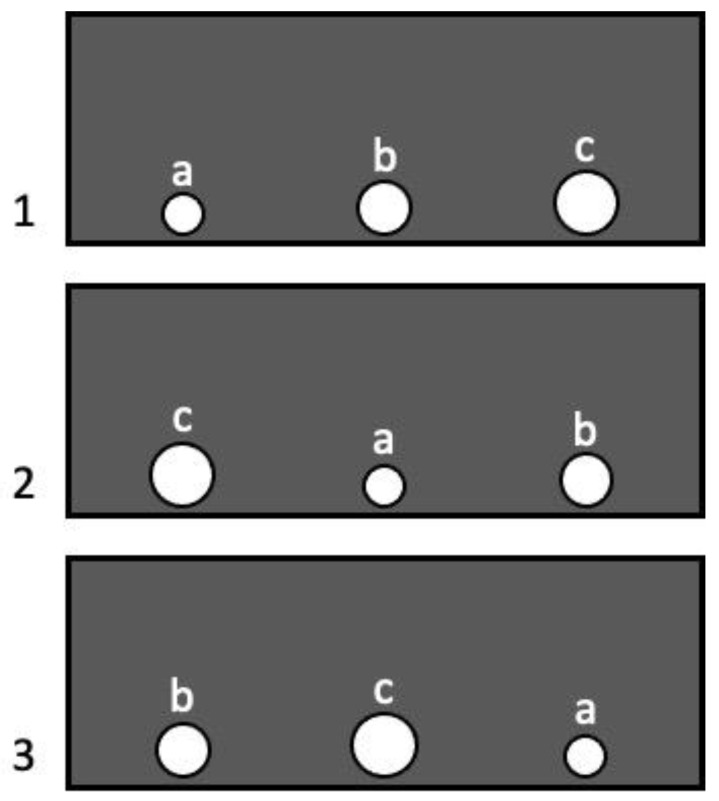
Location of holes during Experiment 1: 1, 2, 3—experimental trials; a—small hole (Ø 70 mm), b—medium hole (Ø 90 mm), c—large hole (Ø 110 mm).

**Figure 4 animals-13-00444-f004:**
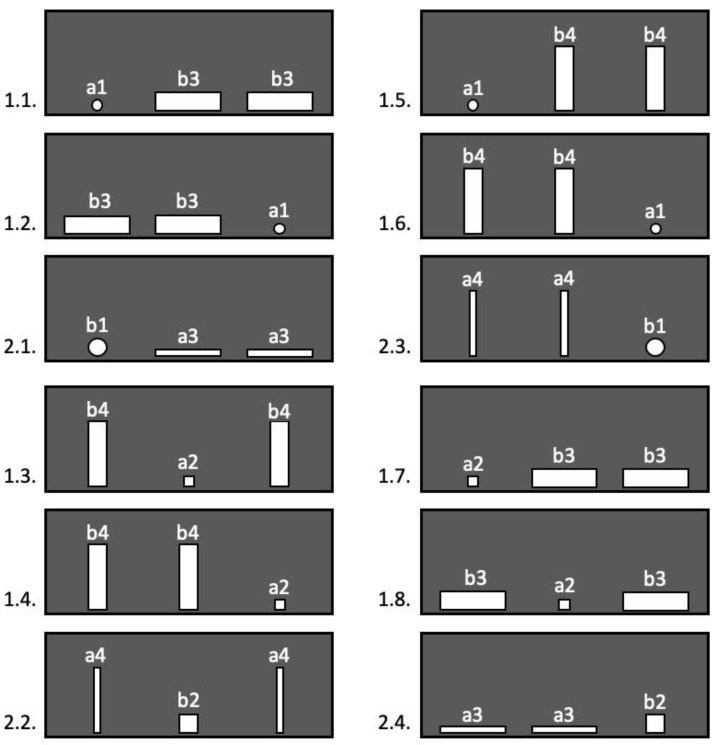
Location of holes during Experiment 2: 1.1.–1.8.—successive background trials, 2.1.–2.4.—successive test trials; a1—impassable hole (Ø 40 mm), a2—impassable hole with a side of 40 mm, a3—impassable hole (horizontal rectangles 250 × 25 mm), a4—impassable hole (vertical rectangles 25 × 250 mm), b1—passable hole (Ø 70 mm), b2—passable hole (square 70 mm), b3—passable hole (horizontal rectangle 250 × 70 mm), b4—passable hole (vertical rectangle 70 × 250 mm).

**Figure 5 animals-13-00444-f005:**
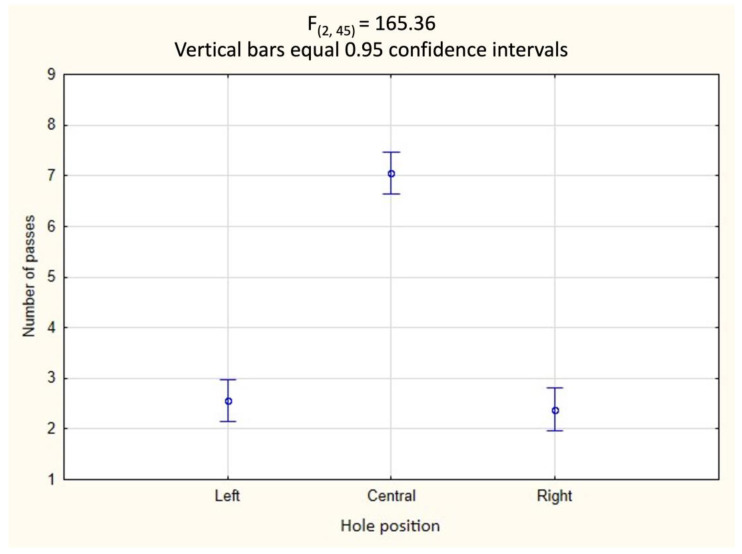
Experiment 1 (*n* = 6): differences in the average number of passages through holes depending on their location in the partition; left, center, or right.

**Figure 6 animals-13-00444-f006:**
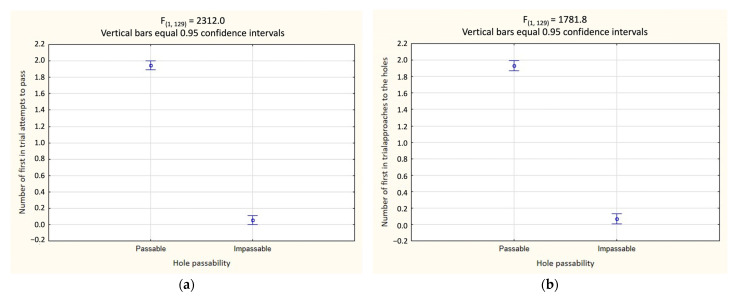
Experiment 2 (*n* = 6): (**a**) differences in the average number of first penetration attempts on the holes as a function of their passability/impassability; (**b**) differences in the average number of first approaches in relation to holes in a trial depending on their passability/impassability.

**Table 1 animals-13-00444-t001:** Correlations between the predictors and dependent variables revealed by factorial ANOVA (significant only).

Predictor	SS	df	MS	F	*p*
Experiment 1 *					
Opening position	252.333	2	126.1667	165.364	0.0001
Experiment 2 **: number of first attempt to pass through the opening					
Passability of the opening	128.,444	1	128.4444	2312	0.0001
Experiment 2 **: number of first approaches related to the opening					
Passability of the opening	124.694	1	124.6944	1781.788	0.0001

* Experiment 1: analysis of 36 test trials; the number of passes is the dependent variable, the opening position is the predictor. ** Experiment 2: analysis of 24 test trials; the dependent variables are the number of first attempts to pass through an opening in a trial and the number of first approaches related to an opening in a trial, the opening passability is the predictor.

## Data Availability

To obtain the data, please contact the corresponding authors.

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
