# Peer review of "Ferrets (Mustela furo) Are Aware of Their Dimensions"

_animals, 2023, doi:10.3390/ani13030444_

Round 1
Reviewer 1 Report (Previous Reviewer 1)
Thank you for making the appropriate changes in the manuscript.
Author Response
Dear Reviwer! The authors are grateful to you for reading and commenting on the article. Authors.Reviewer 2 Report (New Reviewer)
This is an interesting paper assessing body-awareness in ferrets. Exp. 1 is well designed and crucial to establish a baseline for interpreting the results form Exp. 2. Exp. 2 provides a good attempt to test body-awareness in this species, with clear and insightful result.
Whereas I found the manuscript interesting and scientifically sound, I have some comments/doubts on some passages. Please find detailed comments below.
Introduction:
Lines 62 – 70: This passage is a bit ambiguous and would benefit from a re-writing. It is not clear whether it is describing the previous study with crows (not the modified version employed with the ferrets) or whether this is a description of the actual task employed in this experiment (in which case please correct “birds” with “ferrets”).
Lines 80 – 82: In line with current practices, please refrain from novelty claim, and rather mention previous attempts to study self-awareness in ferrets, even when employing tasks other than body-size awareness. For instance, I recall a study on mirror self-recognition in ferrets (https://link.springer.com/article/10.1007/s10071-021-01523-2).
Subjects: Please specify why you used 6 animals (I imagine it is a convenience sample based on the available subjects, but it is not clear from the text). Was this your initial sample size? Did you exclude any animal (and if so, on which criteria)? Please also add an ethical statement identifying the committee approving the studies and confirming that all experiments conform to the relevant regulatory standards for animal research.
Line 118: Please specify the criteria for concluding data collection (how many trials in total for each subject? How many days of testing?). Even though you mention this information later on in the manuscript, I suggest to specify it already at line 118 to ensure clarity for the reader.
Line 130: Please specify the criteria for considering the training over and moving to test (e.g., number of trials, number of training days, behavioural measures, time to solve the trial...)
Line 166: Please clarify why Exp. 1 consisted of 36 trials, while Exp. 2 consisted of 24 test trials and 48 background trials. Please justify this difference.
Methods Exp. 2: Please clarify why you decided to use both round and squared passages (if I am not mistaken, in the original work on hooded crows there were only squared passages).
Results Exp. 2: Did you control for the possibility that ferrets solved the task using a strategy different from estimating body awareness? In particular, from Figure 4 I understand that the two impassable holes looked the same, while the only different one was the passable one. Is it possible that ferrets quickly learnt a same-different rule and chose the different looking passage? You can easily control for this possibility by analysing the very first trial for each subject, to exclude the possible effect of leaning trough trials.
Lines 340 – 342: I agree with the reasoning that the employed task may be more ecologically relevant for ferrets. Yet, I would not conclude that ferrets’ better performance compared to crows suggests that ferrets have a better development of body size awareness. If the task favours one species over the other, it means that the two performances cannot be directly compared. It can be the case of crows having similar capabilities in estimating their body-size, but these were masked by a task that hindered them in expressing such a cognitive trait. As such, I would not draw conclusions about the different degrees of development of body size awareness, but rather reflect on the results of the current study and on the implications of different species-specific research methods.
379 – 381: The list of animals showing instances of body-awareness is not exhaustive (as also mentioned in the discussion, there are attempts to study body-awareness in other species, e.g., in hermit crabs). I would recommend mitigating this last sentence, clarifying that the Authors are mentioning some instances of animals that showed this fine cognitive ability.
Author Response
Dear Reviewer,
Thank you for thorough reading and thoughtful remarks on the article.
Taking into consideration most of them, we have modified the text as follows:
Introduction:
Lines 62 – 70: This passage is a bit ambiguous and would benefit from a re-writing. It is not clear whether it is describing the previous study with crows (not the modified version employed with the ferrets) or whether this is a description of the actual task employed in this experiment (in which case please correct “birds” with “ferrets”).
The passage describes the crow study, so we added one sentence clarifying its meaning (line 62, current version with authors’ corrections, as all line indications hereinafter)
Lines 80 – 82: In line with current practices, please refrain from novelty claim, and rather mention previous attempts to study self-awareness in ferrets, even when employing tasks other than body-size awareness. For instance, I recall a study on mirror self-recognition in ferrets (https://link.springer.com/article/10.1007/s10071-021-01523-2).
We added to the text the study you recommend (lines 82-84).
We are aware of MSR studies of self-awareness, and this type of research was mentioned in an earlier version of the article as well as some other. But, following the recommendation of one earlier reviewer to go straight to the point, which seemed reasonable, we excluded all other methods of studying body awareness but the own body size awareness based on hole penetration method.
Subjects: Please specify why you used 6 animals (I imagine it is a convenience sample based on the available subjects, but it is not clear from the text). Was this your initial sample size? Did you exclude any animal (and if so, on which criteria)? Please also add an ethical statement identifying the committee approving the studies and confirming that all experiments conform to the relevant regulatory standards for animal research.
Information added to the text (lines 89-96).
Line 118: Please specify the criteria for concluding data collection (how many trials in total for each subject? How many days of testing?). Even though you mention this information later on in the manuscript, I suggest to specify it already at line 118 to ensure clarity for the reader.
We have added a more detailed description of the procedure here (lines 148-151).
Line 130: Please specify the criteria for considering the training over and moving to test (e.g., number of trials, number of training days, behavioural measures, time to solve the trial...)
Specifying phrases added (lines 162-165).
Line 166: Please clarify why Exp. 1 consisted of 36 trials, while Exp. 2 consisted of 24 test trials and 48 background trials. Pleasejustify this difference.
Clarifying sentence has been added to the description of Exp.1 (lines 189-191, c.v.). In the description of Exp. 2 the ratio of background and test trials is specified in lines 262-272, c.v. In our opinion, the information is sufficient for a reader not only to understand, but also to replicate the experiments.
Methods Exp. 2: Please clarify why you decided to use both round and squared passages (if I am not mistaken, in the original work on hooded crows there were only squared passages).
An explanatory passage added (lines 262-272).
Results Exp. 2: Did you control for the possibility that ferrets solved the task using a strategy different from estimating body awareness? In particular, from Figure 4 I understand that the two impassable holes looked the same, while the only different one was the passable one. Is it possible that ferrets quickly learnt a same-different rule and chose the different looking passage? You can easily control for this possibility by analysing the very first trial for each subject, to exclude the possible effect of leaning trough trials.
An explanatory passage has been added (lines 420-425)
Lines 340 – 342: I agree with the reasoning that the employed task may be more ecologically relevant for ferrets. Yet, I would not conclude that ferrets’ better performance compared to crows suggests that ferrets have a better development of body size awareness. If the task favours one species over the other, it means that the two performances cannot be directly compared. It can be the case of crows having similar capabilities in estimating their body-size, but these were masked by a task that hindered them in expressing such a cognitive trait. As such, I would not draw conclusions about the different degrees of development of body size awareness, but rather reflect on the results of the current study and on the implications of different species-specific research methods.
We have added some reflections on the conclusions and generally on the comparability of results of cross-species research (lines 447-457).
This is a most important question. One our previous reviewer also reacted to our doubts in possibility of direct comparison of ferret and human child performance in hole penetration tasks, however, that reviewer supported the idea of comparison “in some way”.
For many years we have been studying body/self-awareness/consciousness and accumulated a lot of experimental data on many species: humans, dogs, cats, rats, reptiles (snakes and lizards), invertebrates (snails, cockroaches), etc. With minor exceptions, most our articles were published in Russian. In these studies, we used several paradigms including MSR, hole penetration, body size modification (“own body as an obstacle”), and “feeble support” method (in the studies of body weight awareness). Of course, we are aware of possible doubts in the admissibility of cross-species comparisons. A species-specific method puts another problem, that is, the comparability of methods that differ by definition. And so on. However, this is a question that is not the purpose of this article, and we are open to the discussion elsewhere.
Given this, we make a shortcut and use rather a loose but transparent criterion for this article. Hole penetration is hole penetration, in any case. And to the extent the method is indicative of body size awareness, the results of similar studies conducted on different species can be compared. It is in this precise aspect that ferrets are smarter than crows.
379 – 381: The list of animals showing instances of body-awareness is not exhaustive (as also mentioned in the discussion, there are attempts to study body-awareness in other species, e.g., in hermit crabs). I would recommend mitigating this last sentence, clarifying that the Authors are mentioning some instances of animals that showed this fine cognitive ability.
We have changed the wording of the paragraph in line with the recommendations (lines 518-523).
Once again, thank you for your kind attention to this work and a chance for fruitful (hopefully) discussion.
Authors.

Round 2
Reviewer 2 Report (New Reviewer)
The Authors have addressed all my previous suggestions, and provided thoughtful responses to my comments.
This manuscript is a resubmission of an earlier submission. The following is a list of the peer review reports and author responses from that submission.
Round 1
Reviewer 1 Report
Summary
The question of the study was whether ferrets (Mustela furo ) possess body-awareness, specifically body size awareness. The authors designed a two part experimental study in which the ferrets had to choose one opening out of three to get to bait. The first experiment aimed to find out whether ferrets preferred passages based on size. The second experiment assessed the ferrets’ ability to select a passable hole on the first try while avoiding the unpassable albeit larger ones. Authors found that the ferrets significantly more often choose the middle passage hole to get to the bait in experiment one and in experiment two all tested ferrets preferred to pass through a passable hole on the first attempt. Authors argue that these data indicate that ferrets possess a body size awareness.
Please find a detailled review attached

Author Response
We thank all the reviewers for kind attention to our work.
We have edited the text in accordance with most of Reviewer 1 comments and recommendations (please, use the revised PDF file).
11: its own…
Corrected: one’s own (everywhere)
13/14: indicating the reasonableness of our behavior. What is the meaning of this sentence?
Corrected: indicating the meaningfulness of human behavior (Line 15)
18/19: During the experiment, to reach the bait, the animals had to overcome an obstacle by passing through the holes, some of which were too small for their bodies to penetrate, and some were suitable. Please improve syntax
Simplified: During the experiments the animals had to pass through the holes, some of which were too small for their bodies to penetrate and some were suitable. (Line 19)
20/21: …that ferrets could pre-select the penetrable opening before tactile contact with it, even if the impassable holes were larger – unclear meaning, bad structure
Corrected: …that ferrets could pre-select the passable opening even when the impassable ones were larger in area (Line 21)
27/28: It is known that for ferrets Mustela furo the task requiring penetration into various holes is ecological. And??
Corrected: For ferrets Mustela furo the task requiring penetration into various holes is ecologically relevant. (Lines 28-29)
Further description is in the newly added fragment explaining the choice of the species.
31/32: hole on the first try while the impassable ones were larger in size. Unclear sentence – so why should they not pass through the larger ones?
Corrected: hole on the first try while the impassable ones were larger in area. (Lines 32-33)
32/33: We found that when choosing from the three passable holes the animals preferred those through which the shortest path to the bait ran, the size of the holes was ignored. Is that experiment1 or 2? Bad syntax
Sentence modified: The first experiment showed that when choosing from the three passable holes the animals preferred those through which the shortest path to the bait ran and ignored the size of the holes. (Lines 33-35)
33/34: In the second experiment all tested ferrets preferred to penetrate a passable hole on the first attempt, even though the two impenetrable ones were larger. Unclear meaning. So was it THE passable hole (one out of three) and the other two were larger but not passable? Why not if the were larger?
Because some openings we used were rectangular. A rectangular opening may have a larger area, but either its width or height may be smaller than the least passable dimension.
Corrected: In the second experiment all tested ferrets preferred to penetrate the passable hole on the first attempt, even though the area of the two impenetrable ones was larger. (Lines 35-37)
43: presence of ideas about – rephrase
Corrected: presence of representation of (Line 44)
415: is rare in the animal kingdom – is it rare or just not been studies yet? It has already been found in a number of animals.
Modified: has been studied in only a few non-human animal species. (Line 80)
48: the mirror self-recognition (MSR) test…: If you mention it explain in short what it is.
A brief description of MSR added. (Lines 83-86)
49: other humanoid primates passed this test – non-human primates
Corrected as proposed. (Line 93)
56: both animals and humans…- humans are animals
Corrected: both non-human animals and humans (Line 100)
66: and perceive their body as an obstacle to solving various problems. Meaning? Rephrase
Corrected: and perceive their body as something that hinders solving various problems. (Line 136)
72: previously considered devoid of this ability. Source
Deleted.
72/73: unclear sentence – make no sense in connection with the prior sentence.
Substitution: a description.
93: …an error in only once. Incorrect English
Corrected. This is not English. This is a Copy-Paste residue )) Corrected. (Line 188)
100: also budgerigars, Melopsittacus undulatus, Schiffner, I., Vo, H.D., Bhagavatula, P.S. et al. Minding the gap: in-flight body awareness in birds. Front Zool 11, 64 (2014). https://doi.org/10.1186/s12983-014-0064-y
Added (Line 195, 220-226)
106/107: gender language
As in the source. A kind of impersonal pronoun “he”, “him” and “himself” etc. here. As in “mankind”, in which the part “man-” has nothing to do with masculine gender (a synonym of “human”).
153: incorrect language
Deleted the whole sentence.
157: nutrition, toxicology, pharmacology : source
Sources added (Line 282)
168/170: please improve sentence
Done.
208: minutes
234: An approach was considered a situation ..- unclear sentence
Modified (Line 514)
250/252: rephrase, unclear
Done.
The section is somewhat convoluted, please try to separate and describe more precisely the individual steps and procedures to enable the reader to follow more easily
Here and elsewhere considerable revision and additions are made. (Lines 407-468)
Please rephrase Table headers
Done.
Explain abbreviations used
Following revision, no abbreviations remained.
Please rephrase Figures6 ff: or the number of first in trials – language
Modified.

Reviewer 2 Report
In this manuscript the authors test size awareness in ferrets. This has been previously studied with similar methods in other species.
The introduction is way too long and out of focus. There is no need to expand so much in self awareness, as body size awareness is probably only a component of it (if it has anything to do with it at all).
The capacity to choose openings of appropriate sizes is probably a very widespread and phylogenetically ancient trait, as it is hard to imagine survival without it in most species.
An important detail of the method is missing: what was the relative difference between the body size of the animals and the passable and non passable holes? This is important to determine how a virus it was that the openings were too small or just enough big.
MINOR COMMENTS:
Line 28: ecological should be changed to ecologically relevant
Line 37: why is "mirror self recognition" among the key words? This work has nothing to do with it.
Line 43: I doubt that the presence of "ideas" has been shown in ferrets (or other non-human animals...)
Line 59: the extant literature was not aimed at studying manifestations of self-awareness. Eventually it was aimed at studying some cognitive traits that may be component of what - in the case of humans - evolved in the more complex and fully fledged "self awareness" (see e.g. Fugazza, C., Pongrácz, P., Pogány, A., Lenkei R., Miklósi, A. (2020): Mental representation and episodic-like memory of own actions in dogs. Scientific Reports, 10:10449 DOI: 10.1038/s41598-020-67302-0)
Line 153: change "worse" to "less"
Line 161: What is "The degree of development of ideas"???
Line 169-172: This part should be before in the introduction, explaining why this task was chosen and proividing examples of studies carried out in the wild, showing why this is an achologically relevant task.
Also, an explanation of why ferrets are chosen as a subject species here is missing.
Lines 175-177: In the introduction it is written that the animals were pets. Here it says they were kept in a Centre for research. Were they captive laboratory animals or pets?
Why is not the analysis of the background trials provided?
Line 335: change "sample" with "trials"
Line 380: This argument is not understandable to me. It should be explained better.
Lines 401-402: The results of this study are not comparable to results of previous studies because the methods differ. This sentence should be deleted or riformulated
Line 431-433: Again, the studies on human infants are not comparable to this one. This sentence should be deleted.
Line 434: "environmentally friendly" should be changed to "ecologically relevant"
Lines 445-448: Body size awareness is not one of the manifestations of self-awareness. The results of the study should not be over-interpreted.
I suggest a careful revision of the introduction and conclusion, deleting all the over-interpretations of the results.
Author Response
We thank all the reviewers for kind attention to our work.
We have edited the text in accordance with most of Reviewer 1 comments and recommendations (please, use the revised PDF file).
In this manuscript the authors test size awareness in ferrets. This has been previously studied with similar methods in other species.
The introduction is way too long and out of focus. There is no need to expand so much in self awareness, as body size awareness is probably only a component of it (if it has anything to do with it at all).
Reconsidered, modified and somewhat reduced.
The capacity to choose openings of appropriate sizes is probably a very widespread and phylogenetically ancient trait, as it is hard to imagine survival without it in most species.
Most probably. Some considerations added.
An important detail of the method is missing: what was the relative difference between the body size of the animals and the passable and non passable holes? This is important to determine how a virus it was that the openings were too small or just enough big.
Data and description added. (Lines 534-536)
MINOR COMMENTS:
Line 28: ecological should be changed to ecologically relevant
Corrected. (Line 29)
Line 37: why is "mirror self recognition" among the key words? This work has nothing to do with it.
Deleted.
Line 43: I doubt that the presence of "ideas" has been shown in ferrets (or other non-human animals...)
Modified: representation (Line 100)
Line 59: the extant literature was not aimed at studying manifestations of self-awareness. Eventually it was aimed at studying some cognitive traits that may be component of what - in the case of humans - evolved in the more complex and fully fledged "self awareness" (see e.g. Fugazza, C., Pongrácz, P., Pogány, A., Lenkei R., Miklósi, A. (2020): Mental representation and episodic-like memory of own actions in dogs. Scientific Reports, 10:10449 DOI: 10.1038/s41598-020-67302-0)
Added.(Line 109 and References)
Line 153: change "worse" to "less"
The whole phrase was modified.
Line 161: What is "The degree of development of ideas"???
Changed to: development of representations (Line 286)
Line 169-172: This part should be before in the introduction, explaining why this task was chosen and proividing examples of studies carried out in the wild, showing why this is an achologically relevant task.
Also, an explanation of why ferrets are chosen as a subject species here is missing.
Explanations added.
Lines 175-177: In the introduction it is written that the animals were pets. Here it says they were kept in a Centre for research. Were they captive laboratory animals or pets?
Mention of pets in the introduction has been deleted. Pet puppies were raised in the lab. So they were laboratory animals.
Why is not the analysis of the background trials provided?
Added, (Lines 454 and further, 628-636)
Line 335: change "sample" with "trials"
Changed everywhere.
Line 380: This argument is not understandable to me. It should be explained better.
Modified.
Lines 401-402: The results of this study are not comparable to results of previous studies because the methods differ. This sentence should be deleted or reformulated
Deleted.
Line 431-433: Again, the studies on human infants are not comparable to this one. This sentence should be deleted.
Revised.
Line 434: "environmentally friendly" should be changed to "ecologically relevant"
Corrected (Line 29, 659)
Lines 445-448: Body size awareness is not one of the manifestations of self-awareness. The results of the study should not be over-interpreted.
We introduced (hopefully) a more careful wording. (Lines 770-777)
I suggest a careful revision of the introduction and conclusion, deleting all the over-interpretations of the results.
Done.

Round 2
Reviewer 1 Report
please find all comments and mark ups in the attached document

Author Response
Dear Reviewer
Thank you for thorough reading of our manuscript and comments.
We apologize for some new errors in the text that seem to appear while editing, such as chunks of deleted words and sentences, doubling articles and prepositions, etc. It is our hope there won’t be many (any?) this time.
Also thank you for some friendly comments, specifically the one regarding the interspecific comparability of experimental results. From our experience in experimental work, we understand very well that it is next to impossible to strictly reproduce either the methods or the results obtained, say, in human children across many other mammal species, to say nothing of birds, reptiles or crabs. Yet they may be compared at a certain level of abstraction, and some illuminating and fruitful ideas may arise from such a comparison.
We attach the file, that contains all latest corrections.
Best,
Authors

Round 3
Author Response
Dear Editors and Reviewers,
We have uploaded the latest revised version of the manuscript on ferrets’ body size awareness.
The corrections were made either by strictly following the reviewer’s recommendations or, in some specific cases, as we think appropriate. We also added Latin denominations to all English denominations of the animal species when mentioned for the first time. Some highlighted phrases are deleted or almost totally reformulated.
Several minor additional corrections were made to avoid possible ambiguity.
Thank you for thorough reading of the manuscript and your thoughtful remarks.
Best,
Authors.
